# Infrastructuring the Circular Economy

**André Nogueira [1],\*, Weslynne Ashton [2] , Carlos Teixeira [3], Elizabeth Lyon [4] and Jonathan Pereira [4]**

[1]  Harvard T.H. Chan School of Public Health, Harvard University, 677 Huntington Avenue, Boston, MA 02115, USA

[2]  Stuart School of Business, Illinois Institute of Technology, 10 West 35th Street, Chicago, IL 60616, USA; washton@iit.edu

[3]  Institute of Design, Illinois Institute of Technology, 10 West 35th Street, Chicago, IL 60616, USA; carlos@id.iit.edu

[4]  Plant Chicago, 4459 South Marshfield Avenue, Chicago, IL 60609, USA; elizabeth@plantchicago.org (E.L.); jonathan@plantchicago.org (J.P.)

\*  Correspondence: anogueira@hsph.harvard.edu

**Abstract:** The circular economy (CE), and its focus on the cycling and regeneration of resources, necessitates both a reconfiguration of existing infrastructures and the creation of new infrastructures to facilitate these flows. In urban settings, CE is being realized at multiple levels, from within individual organizations to across peri-urban landscapes. While most attention in CE research and practice focuses on organizations, the scale and impact of many such efforts are limited because they fail to account for the diversity of resources, needs, and power structures across cities, consequently missing opportunities for adopting a more effective and inclusive CE. Reconfiguring hard infrastructures is necessary for material resource cycling, but intervening in soft infrastructures is also needed to enable more inclusive decision-making processes to activate these flows. Utilizing participatory action research methods at the intersection of industrial ecology and design, we developed a new framework and a model for considering and allocating the variety of resources that organizations utilize when creating value for themselves, society, and the planet. We use design prototyping methods to synthesize distributed knowledge and co-create hard and soft infrastructures in a multi-level case study focused on urban food producers and farmers markets from the City of Chicago. We discuss generalized lessons for "infrastructuring" the circular economy to bridge niche-level successes with larger system-level changes in cities.

**Keywords:** circular economy; design; industrial ecology; infrastructure; participatory action research; socio-ecological-technical systems

## 1. Introduction

The circular economy (CE) has captured the imagination of civil society and corporate actors as a framework for realizing the elusive goals of sustainability [1]. It has presented a tangible way to contribute to the sustainability of resource flows and allocation in contemporary production and consumption systems, by focusing on improving the effectiveness of current management practices in organizations through extending the useful life of products, and sharing, recycling, and regenerating resources [2].

Cities face urgent needs to expand and advance CE adoption and sustainability-oriented practices due to growing concerns over climate change, environmental pollution, and the inequitable distribution and allocation of resources in the linear (take-make-waste) economy [3,4]. They present unique opportunities for CE interventions as they are territories where human populations are concentrated,

and where multiple natural (ecological) and man-made (social and technical) systems intersect, diverse human and non-human agents interact, and different types of resources are created, transformed, circulated, used, and wasted [5]. Cities are complex socio-ecological-technical systems (SETS) with nested subsystems such as neighborhoods, organizations, and infrastructure networks, and are themselves part of larger SETS such as states and nations.

In cities, social, ecological, and technical systems are deeply intertwined since the parts of one subsystem belong to and dynamically affect the others. Urban infrastructures are key elements that integrate these systems and are thus responsible for the interconnectivity and interdependency of resource flows and allocation across their levels; for example, policy and strategies at the macro-level, organizations and operations at the meso-level, and patterns of daily life and related tactics at the micro-level [6].

## 1.1. Urban Infrastructures

Modern urban infrastructures have become local nodes of global operations, and the means through which individuals and organizations can (or cannot) access and mobilize resources that traverse local, regional, and global sub-systems. When modern urban infrastructures were built, they did not anticipate the multitude of resources that would flow through them, the speed at which these flows would travel, and the complex inter-linked ramifications they would promote across social, ecological, and technical subsystems. Most of them were predicated on the goals of economic growth and progress, designed through the technical lenses of health and safety, and informed by principles of functionality and efficiency [7]. Yet, they failed to account for the negative and positive feedback created as resources were mobilized at faster rates than could be assimilated within urban and larger-scale ecosystems across the globe [8,9]. Intervening towards sustainability through the narrow focus of circular flows of resources among organizations ignores the many other flows that these organizations are trafficking (e.g., political agendas, cultural preferences), which have an equal or greater impact on how well-being is experienced locally [10].

Urban food distribution infrastructure in the United States (US), for example, was designed to aggregate commodities produced on large-scale, industrial farms, and distribute them to populations across the nation [11]. While these infrastructures enabled economies of scale and reduced the prices paid by consumers, they supported the generation of significant amounts of waste and accelerated environmental degradation along the entire value chain at an unprecedented speed. They also privileged economic growth and profitability of large producers over the development of local economies [11–14]. Thus, small producers, particularly ethnic minority-owned businesses in urban and peri-urban areas in the US, face challenges with access to capital and formally entering the food supply chain [12]. Environmental challenges promoted by the use of modern infrastructures include high energy and water use during production, large carbon footprints in transportation, and large volumes of waste throughout its stages [14,15]. Equally challenging are aspects related to social and economic dimensions because low income and ethnic minority populations often face disproportionate health burdens as they lack access to healthy, nutritious food, due to low availability and affordability of such food in their neighborhoods [16,17]. The food distribution infrastructure is representative of how modern infrastructures became the pathways for the circulation of many types of resources (e.g., money, knowledge, power, etc.) that were disconnected from the dynamic needs and interactions of local populations. This food production and distribution scenario reflects the complex challenges found in many urban SETS, in which multiple subsystems and variables interact, producing both desirable and undesirable emergent outcomes and feedback among the system components that can become entrenched and difficult to disrupt and change [18].

Modern infrastructures were conceptualized and built to last decades, if not centuries. They consolidated and perpetuated unsustainable and inequitable patterns of flows of resources within and across technical systems, directly influencing the well-being of people, organizations, and the natural environment (e.g., public safety policies, mobility systems, remediation processes, etc.). Over

time, knowledge about how to intervene in infrastructures evolved with a distinction between two dimensions. Hard infrastructures relate to tangible and material aspects. It usually results from engineering and natural science design efforts and includes mostly technological elements, such as products and their mechanisms of operation. Soft infrastructures relate to institutions, intangible aspects, and social behavior. They are centered on the exploration of human interactions, services, and networks, and therefore, consider multiple perspectives towards unfolding new dynamics in systems [19,20].

The discussion between hard and soft dimensions reflects the paradoxical and unruly nature of infrastructures. According to Larkin, "the duality of infrastructures indicates that when they operate systemically they cannot be theorized in terms of the object alone" [21] (p. 329). On the one hand, they symbolically represent the idea of the commons, and the possibility of access to resources and assets that facilitate everyday life. On the other hand, they implicitly condition everyday life through their design, consequently reinforcing established power dynamics in the circulation and allocation of resources. Hard and soft infrastructures connect social networks by providing access to different types of resources and shaping the context for how people can or cannot work, learn, play, and live with others. As they are considered mature elements upon which activities of daily life and environmental performance fundamentally depend [22], the existing infrastructures shaping production and consumption systems are usually perceived as given and unchangeable. This is one of the reasons why contemporary CE practices are constrained by them.

## 1.2. Urban Infrastructures and CE

CE initiatives at the city-scale can benefit from theoretical debates around the role of infrastructure in shaping social, ecological, and technical dynamics [23–26]. In their studies, Star and Ruhleder recognized that properly working infrastructures are formed considering a set of standards and protocols of both its soft and hard dimensions [27]. According to the authors, once built, infrastructures carry a system of offerings (e.g., people, objects, environments, messages, and services) and affordances that standardize the circulation and allocation of resources, as well as how the infrastructure is used. Instead of approaching infrastructure as an element "which runs underneath actual structures", they suggested individuals and organizations recognize them as relational elements "upon which something else rides, or works, a platform of sorts" [27] (p. 151). Such an approach is particularly useful for CE initiatives happening within cities, where new technologies and new dynamics of daily life are rapidly changing and the fairly stable, technical elements of the 20th-century infrastructure are posing significant barriers to progress towards overcoming 21st-century sustainability and equity challenges.

Urban change agents are searching for creative alternatives for CE practices, business models, and new offerings that have higher "fitness" between local socio–ecological dynamics and the technical capabilities necessary to actualize a CE [28,29]. These include citizen-led material reuse and recycling centers, such as La Recyclerie in Paris, France [30] and Recycle Here in Detroit, Michigan [31], as well as local government-led CE initiatives, such as in Amsterdam, Netherlands [32] and Charlotte, North Carolina [33]. However, many CE practices within urban environments often remain novelties at the meso- or niche-level and are unable to scale as they attempt to activate and mobilize multiple resources through pathways that counter the linear logic underlying the design of these infrastructures.

Furthermore, many individuals and organizations who are exploring large-scale CE urban interventions operate within their own traditional disciplinary silos, leverage existing networks of partners, or do not involve residents and local organizations in their processes. As a result, these interventions tend to be led, funded, and implemented by a small set of agents that lack expertise about the dynamics of the daily life of residents and practices of local organizations [34,35]. Without expanding their scope of closing loops in material resources from existing organizational practices, CE interventions at the city-level will continue to fall short in understanding how urban dynamics are shaped by a much more complex web of interconnected infrastructures responsible for allocating and mobilizing social, ecological, and technical flows of resources.

To effectively realize the CE in an urban context, micro-level changes must be able to leverage existing infrastructure or be robust enough to transform them in order to achieve lasting structural change. Likewise, macro-level interventions at the city scale need to reflect the challenges faced by the individuals and organizations operating within their urban boundaries. Ozanne and Saatcioglu suggest that in order to succeed in promoting paradigm shifts in complex systems, interventions "must occur at multiple levels and depend upon a considerable investment of resources" [36]. Similarly, Klerkx and colleagues argued that bottom-up approaches often obscure other influential factors, such as technological advancements and new institutional arrangements that are present at different levels [37]. For them "innovation requires work on changing relationships and institutions at different levels" [37]. This demands that agents embedded in a particular context understand and engage in dynamics beyond the interactions happening on the focal level. New pathways are required to mediate the complementary roles of top-down strategic approaches and bottom-up emergent transformations [38].

The concept of "infrastructuring", from participatory design, suggests an expansion from focusing on the hard dimensions of interventions (outputs), and to also consider people's activities and organizational practices (processes) [39]. It considers that processes for designing interventions, which determine the allocation and circulation of resources, are just as relevant for the CE as the outputs of these processes, or the new (hard) elements through which resources circulate. Thus, "infrastructuring" CE presents a means to democratize the processes of determining how resources should be allocated and mobilized. It requires the participation of individuals and organizations to be involved in and impacted by new infrastructural interventions during the processes of creating them, not only in the implementation phase. Such an approach considers the multiple agents' aspirations and challenges as input to determine the goals of infrastructural interventions, the new offerings enabled by them, and the actions these offerings afford users to do. Without "infrastructuring" CE, it is likely that contemporary interventions will continue to fall short in recognizing how both the hard and soft dimensions of infrastructures determine the circulation and allocation of resources, and therefore the sustainability and equity of these infrastructures.

### 1.3. Study Objectives

There is a significant gap between the aspirations of a CE, the strategic approaches and operational capacities of local organizations, and the understanding of the dynamics of daily lives of diverse urban residents. This research spans multiple levels and systems across the City of Chicago where knowledge in systems thinking (ST), industrial ecology (IE), and design (D) was combined to explore pathways through which the narrow focus of CE initiatives on material resources at the meso- (organizational) level could be expanded.

This paper presents the results of one of these projects: a collaboration between Illinois Institute of Technology (IIT) and Plant Chicago (PC), a not-for-profit organization that was created to promote research and education activities at The Plant, a community of sustainable urban agriculture and food businesses, co-located in a former meatpacking factory. The research team co-created a framework to incorporate considerations of multiple types of resources shaping organizational practices and influencing dynamics at the macro-level (e.g., policy and related strategies) and at the micro-level (local patterns of daily life and related tactics). The team also co-developed a model to support multi-level infrastructural interventions that can enable paradigm shifts for actualizing CE at the city-level. PC leveraged both the framework and the model to scale up its impact on the local circular economy both at the facility-level, working with a small network of co-located food production businesses, and at the city-level, infrastructuring a city-wide network of farmers markets.

## 2. Materials and Methods

Individuals and organizations at The Plant have developed collaborative material reuse projects including organic waste through composting, construction materials through careful demolition and reuse, reduction and redesign of packaging, and surplus exchange, among others. The research team

investigated how these collaborative efforts, engagement with diverse sets of agents, and knowledge creation being developed and disseminated at the meso-level could be scaled for greater impact in the City of Chicago.

This research utilized mixed methods with participatory action research (PAR) as the primary methodology for engaging with and involving diverse agents to understand and co-create CE interventions, which was an inherently iterative process. Researchers integrated tools from industrial ecology and systems thinking with design frameworks and methods as a means to both gather and analyze different types of data, as well as support collective decision-making about alternative futures with those embedded in the context of research (see Figure 1). PAR reinforces processes of 'learning by doing' and focuses on creating an action-learning system for developing practical solutions for complex social problems. It actively engages individuals and organizations embedded in the context being researched during all stages of the research process, ensuring greater involvement and collaboration between all parties, including the preparation of activities, collection, analysis, and synthesis of data, and validation of the outputs [40]. PAR considers that existing knowledge and resources distributed across agents are as valuable in the research process as they are in contributing to the outcomes. The standpoint is that those affected by the research should have a say not only in the research outputs and outcomes, but also in the process [41]. Thus, PAR ensures that individuals and institutions have agency in the research being done about the context in which they are embedded [42], while recognizing their multiplicity of voices, values, and concerns.

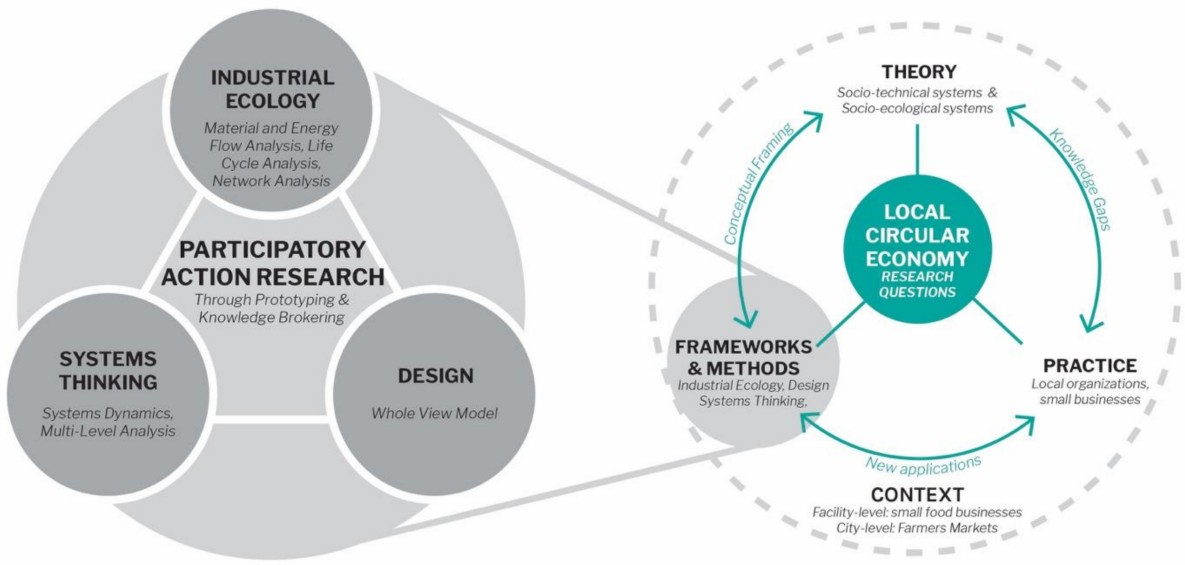

**Figure 1.** Underlying structure of the research approach.

The combination and integration of tools and frameworks throughout PAR activities were made possible through prototyping and knowledge brokering methods. Unlike traditional prototyping approaches that search for refinement of an existing concept, researchers leveraged prototyping methods for knowledge brokering. Both prototyping activities and their related prototypes were used to engage and involve diverse agents, and became the means to (1) discover new information about the context, (2) test hypotheses and concepts (associated with products and knowledge), (3) collect evidence for further analysis and interpretation, and (4) explore alternative futures co-defined by agents who do not interact with one another normally. Through prototyping activities, participant engagement, data gathering, data analysis, and the production of interventions became generative activities that allowed researchers and other agents to explore previously unarticulated (and often hidden) connections and challenges related to CE practices. Upon revealing these challenges, researchers created and prototyped a new framework that expanded the consideration of the resources flowing, as well as a new model

to shape the flows of these resources in activities of infrastructuring CE [43]. Both are presented in this article.

Tools, frameworks, and methods from the three fields (IE, ST, D) were used as artifacts to facilitate and mediate engagements during prototyping activities for both discoveries of system dynamics and refinement of concepts, workshops and focus groups, and other hands-on activities of PAR. Material and energy flow analysis (MEFA) and life cycle assessment (LCA), as well as network analysis, were used to measure and map the flow and interactions of materials, money, and relationships among actors. Conventionally, MEFA is applied to track the inputs, outputs, and transformation of specific material and energy resources through a system of interest, and uses existing business practices, databases, inventories, and surveys as its primary sources for data [44]. In parallel to quantifying the materials consumed and produced by different actors, researchers leveraged IE tools to explore the potential for material reuse and cycling (industrial symbiosis) with each one of the businesses. Combined, these activities allowed for identification of common barriers across organizations concerning data gathering and sharing. Likewise, traditional network analysis maps different types of relationships among actors and quantifies their correlations. However, since researchers involved different business owners and staff in their mapping activities, different perceptions about business interactions and the multiple types of values being exchanged between them were also surfaced.

Systems dynamics modeling helped to surface key variables influencing the interaction among agents and discuss patterns of challenges underlying engaging in CE practices; for example, the nature of the relationships between organizations considering business to business, informal trading of different types of resources, and social interactions [45]. By mapping causal relationships and feedback mechanisms among these variables, researchers and partners explored leverage points of interventions, considering how changes in specific components could affect the whole [46].

Design methods and frameworks from the whole view model [47] were used as a structure for brokering different types of knowledge distributed among diverse agents throughout all phases [48]. These include, but are not limited to, performing field and user observations, and facilitating activities with different tenants and their staff for describing the current state of their organizations and their offerings, as well as to speculate alternative futures. Having a rigorous and pliant structure to perform various activities allowed the research team to better understand the patterns of strategic and operational challenges across businesses located in the facility, as well as patterns of daily life of those working in the building.

Combined, MEFA, LCA, network and systems dynamics maps, and the whole view model helped researchers to co-create a more holistic baseline with multiple agents that resulted not only in better understanding the parts of the SETS that were embedded, but also the various relationships among them through the flow and allocation of different types of resources.

## 3. Results

This research project can be described considering three central foci: (1) quantifying material and energy flows and circular economy potential (facility-level); (2) activating and mobilizing agents for advancing circular economy practices (facility-level); and (3) scaling niche-level CE success to city-level impact (city-level). Although the center of gravity expanded, researchers continued to perform all activities related to previous foci throughout the project. Thus, the results in both levels were generated iteratively alongside framework and model development. That is, the outputs of prior activities led to new questions, for which new frameworks and models were identified or created, which led to new outputs, expanding the scope of the project for the next iteration and expansion. Figure 2 presents an overview of the iterative research processes.

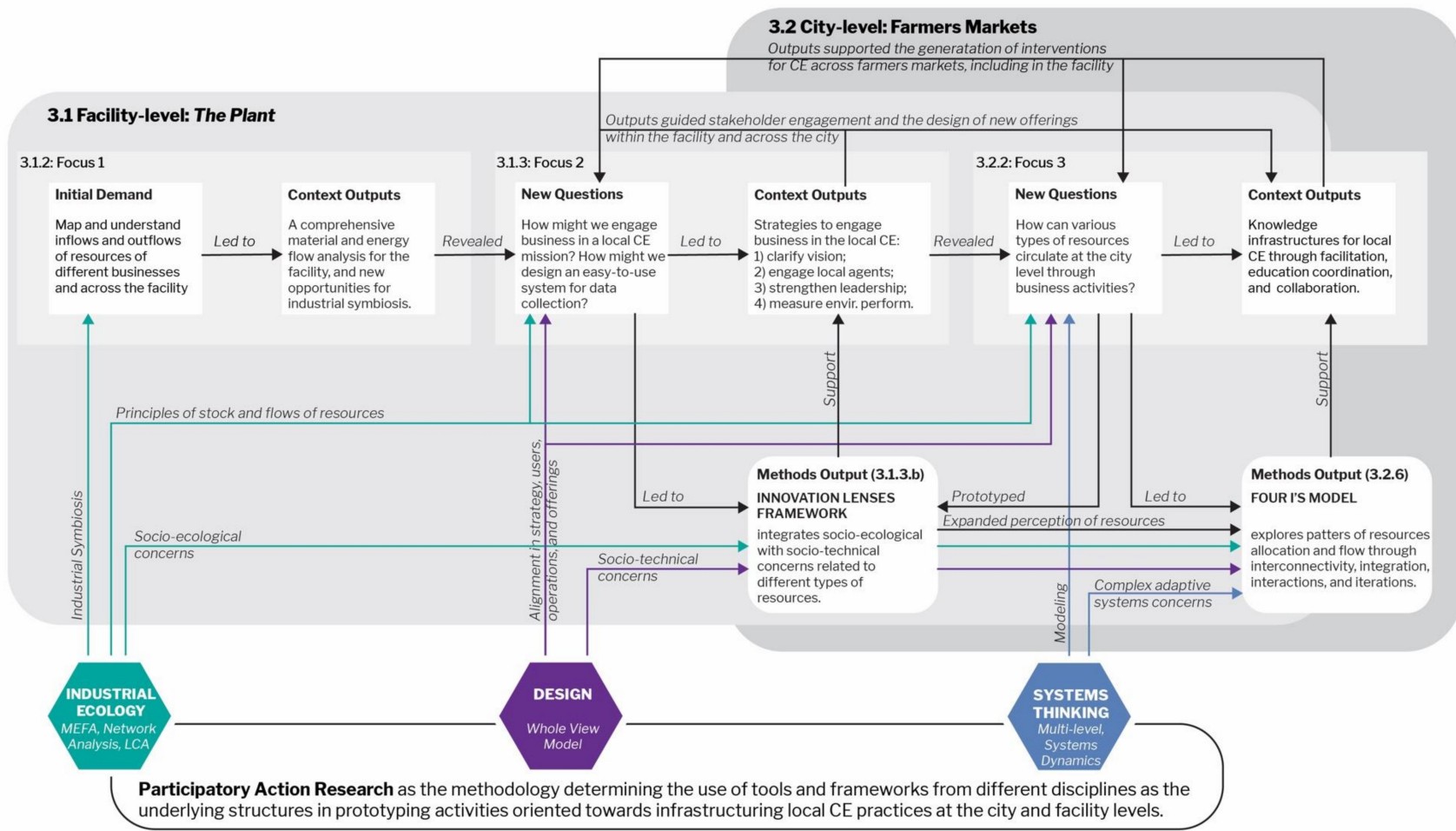

**Figure 2.** Iterative development of methods and results across facility and city-level projects. Abbreviations: CE, circular economy; MEFA, material and energy flow analysis; LCA, life cycle assessment.

*3.1. Facility-Level: The Plant*

3.1.1. Contextual Background and Problem Framing

The Plant, located in the Back of the Yards neighborhood on Chicago's South Side, was started in 2010 by Bubbly Dynamics (BD), a for-profit social enterprise whose mission is "the creation of replicable models for efficiencies that close loops of waste and energy and to encourage others to implement these techniques to combat climate change" [49]. During the first half of the 20th century, Chicago's Back of the Yards district was considered the meatpacking center of the US. The neighborhood grew at the intersection of many railroads and became home to advances in production systems, including the first refrigerated boxcar in 1880, and the country's oldest community organization, Back of the Yards Neighborhood Council (1939). In 1971, the closure of the stockyards was the main trigger for degradation of the neighborhood, including high levels of soil contamination, lack of education, and a high concentration of low-income and marginalized populations.

BD's vision for The Plant was for it to become a model for new sustainable urban food production and consumption systems by transforming an abandoned industrial building into a fully occupied food and beverage business incubator [50]. Over the last decade, The Plant has been home to a collective of urban agriculture and sustainable food start-up or early-stage enterprises, most of them drawn to the facility by its sustainability and circular economy ethos. In 2017–2019, the building housed two dozen tenants including indoor and outdoor farms, kombucha and beer breweries, a cheese distributor, a coffee roaster, and other small food and beverage producers and distributors. Together, they occupied about 80% of the facility and employed 85 full-time employee equivalents. Among the tenants was Plant Chicago (PC), a non-profit formed by BD to lead education and research activities at The Plant, particularly around CE [51]. Among other offerings and programs, PC promoted and operated an on-site farmers market that became a venue for tenants in the building to sell their products and engage with visitors from across the city and around the world, and to a lesser extent, people from the neighborhood.

The renovation of the facility has been gradual. Instead of taking a traditional approach to physical redevelopment, through which high upfront investments determine the quality, allocation, and use of different types of resources, including space design and hard infrastructure within facilities, BD followed a path of adaptive growth. Such an approach resulted in a series of infrastructural interventions that co-evolved in a non-linear manner with the practices they underpin. The company repurposed different parts of the building as new tenants desired to occupy these spaces, or when current tenants requested specific changes based on their operational evolution. This is not only a financial strategy to reduce the uncertainty of resource allocation, as BD uses rents of the finished spaces to reinvest in the building or fund the remodeling of new spaces for incoming tenants, but also an ecological approach informed by principles of complex adaptive systems, such as self-organization and emergence. For example, as new spaces are occupied, additional byproducts become available, giving room for BD to experiment with new technologies (e.g., redirecting carbon dioxide from brewing beer to enhance plant growth in a vertical farm). By having time flexibility and control over the construction process, BD maximizes reuse of the materials in the building.

Overall, the adaptability of The Plant community was shaped by the evolution of interactions between human, technical, and ecological components of the system. As a collaborative, ongoing redevelopment process, the interactions between these elements are key to maintain, augment, or abandon sustainability-oriented practices enacted within the facility. Thus, each infrastructural transformation has built upon and been shaped by previous interventions, including this research and the impacts it has created in the facility.

3.1.2. Focus 1: Quantifying Material Flows, Industrial Symbiosis, and Circular Economy Potential at The Plant

In summer 2016, researchers (W.A., J.P., and other collaborators) conducted the first comprehensive material flow analysis for The Plant, measuring material inputs, electricity use, and waste generation at individual tenant businesses and the facility as a whole [52]. The results of this research highlighted the need for 1) a framework to actively engage small business occupants of the facility in the overall mission of moving towards the circular economy, and 2) an easy-to-use and ongoing system to collect data and measure the sustainability performance of the individual actors and facility as a whole. Follow-up work by BD and tenants focused on building the physical infrastructure for increasing facility-scale industrial symbiosis [50]. For example, inefficient water use was highlighted as a major deficiency in the 2016 study, and BD subsequently designed and installed a rainwater collection system for use by building tenants.

3.1.3. Focus 2: Activating and Mobilizing Agents Advancing Circular Economy at The Plant

PC, BD, and researchers (W.A., A.N., and IIT graduate students) started the current research collaboration to explore sustainability performance measurement frameworks with which the occupants of The Plant could track their material and energy flows. The assumption was that with a framework in place, tenants would engage more in sustainable initiatives because they would have more information regarding the technical and financial viability of engaging in CE practices with The Plant community. Researchers reviewed existing sustainability performance frameworks and reporting tools to determine best practices. In parallel, researchers conducted user and field observations, as well as semi-structured interviews with tenant business owners and staff, BD and PC staff, volunteers, visitors, and several others engaged in The Plant community. These observations revealed key barriers to engaging in and making progress on CE initiatives. They also helped to uncover the priorities of the facility owners, main incentives underlying engagements, and critical challenges concerning individual operations and collective CE efforts. For example, without a formal structure and proper mechanisms to collaborate, current (particularly new) tenants were struggling to understand their role in promoting, leading, or participating in the CE practices proposed by PC and BD.

A series of co-creation workshops were subsequently held with stakeholders within this community to chart a path for deepening and sharing an understanding of CE and the challenges faced by individual companies in enacting it. Collectively, these interactions resulted in the identification of four strategies for achieving this mission, each with its own set of tactics: (1) clarify PC's and BD's vision for CE; (2) engage a broader set of stakeholders; (3) strengthen leadership and support within the facility; (4) measure environmental performance. The staff of BD and PC, as well as some of the tenant business owners within the community, took leadership positions in this endeavor, serving as repositories of technical, business, and tacit knowledge to educate and engage the newer tenants in the facility on the CE journey. A research report was created describing these strategies with tactical recommendations, distributed among tenants and served as the background for the next phase of research [53].

Expanding Perception of the Circular Economy at the Plant

In summer 2017, the researchers (A.N.) built a research base inside The Plant and performed a series of activities that ranged from co-defining CE with tenants, lunch and learn meetings, and design-led workshops with tenants and volunteers, among others. The overall goal was to build new relationships between PC, multiple tenants and their staff, residents of the surrounding neighborhood, volunteers at The Plant, and BD. Researchers used "infrastructuring" to better understand the hard and soft infrastructures conditioning CE practices, as well as the individual activities and business practices that mobilized different types of resources through these infrastructures to actualize CE within the facility. Systems maps were used as physical artifacts to encourage discussion through which different agents in The Plant community could continuously manage concerns and controversies related to implementing CE practices together. In doing so, researchers and participants were able to adapt their

activities and consider the unintended consequences of their actions, even as new concerns emerged throughout these engagements.

Interviews with participants alluded to the use of resources beyond the natural, manufactured, and financial resources, such as cultural norms and personal relationships, to actualize CE. To bridge the gap of what is typically considered in CE research, the team explored more expansive frameworks for assessing how different types of resources are utilized by organizations and identifying new leverage points for overcoming the observed barriers to CE adoption [54].

Innovation Lenses Framework

It was in this context that researchers created the innovation lenses framework and developed a series of prototyping activities to explore how the considerations of social, ecological, and technical concerns could be leveraged to increase the effectiveness of CE practices and broaden participation in them. The framework resulted from integrating socio-ecological and socio-technical concerns, considering eight different types of capital, defined as "any type of resource capable of producing additional resources" [55] (p. 165). The capitals can be classed in three broad categories: social, ecological, and technological (see Table 1). The "social" capitals are created through human activities and interactions, these include human, social, political, and cultural capital. Ecological capital is well-defined for only those natural resources that are deemed valuable and monetized, such as fossil fuels, minerals, and water [46]. The "technical" capitals are human-made resources that are only present in economic activities and include financial, digital, and manufactured resources. The framework has been used to describe how different types of resources are used in existing SETS, and to generate interventions that embrace an expanded understanding of the dynamics shaping CE options.

**Table 1.** Definition of the eight capitals as innovation lenses. Reprint with permission [54]; Copyright 2019, Elsevier.

| Social | | | |
|---|---|---|---|
| **Human** | **Social** | **Cultural** | **Political** |
| The ability and capability of individuals to produce and manage their well-being. It includes individual health, knowledge, skills, and motivation. | The professional and social connections among agents. It includes partnerships and collaborations, as well as informal gatherings. | Values and beliefs inherent in social practices, or incorporated by communities, that determine patterns of behavior being encouraged, discouraged, or tolerated by individuals and organizations over time. It also includes ethnicity, spirituality, heritage, traditions, and daily practices. | Governing structures in organizations that determine how decisions are made and power is distributed. It involves hierarchy, inclusion, equity, transparency, access, and participation. |
| **Ecological** | **Technical** | | |
| **Natural** | **Financial** | **Manufactured** | **Digital** |
| Comprises natural resources, both renewable and nonrenewable. It also includes fauna and flora, as well as their life-supporting systems. | The productive power in the resources of other types of capitals. It includes the resources and assets of an individual or entity translated in the form of a currency that can be accessed, owned, or traded. | All material goods. It includes human-made elements such as physical infrastructures, roads, artifacts, and machines. | Digital infrastructure and data. It includes digital platforms, as well as the mechanisms of data collection, analysis, and storage. |

The application of the framework allowed participants to raise questions that reflected a systemic understanding of how different types of resources flow and are allocated based on the individual's activities and business practices within The Plant (see [54]). Participants explored alternative pathways to activate and mobilize available resources and assets for them to contribute to the CE within and beyond the facility's boundaries. For example, PC realized that the value in the human resources in the surrounding neighborhood had not been tapped because of the focus on material loop closing at The Plant, and so sought to bridge practices at the meso-level in the facility, with the needs of surrounding residents. Awareness of the disconnect between achieving "facility-scale Industrial Symbiosis" and investing in residents of the Back of the Yards led tenants at The Plant, including PC and BD, to prioritize hiring and training their neighbors to work inside the building. They also began to support neighborhood entrepreneurial ventures, both by providing infrastructures for them to develop their offerings and by welcoming small neighborhood businesses in their farmers market. Combined, these activities supported PC staff in realizing that their organization was lacking alignment between its goals to incubate local circular economies, current strategies, and its offerings to actualize CE practices. As a response to this challenge, researchers and PC staff collaborated to build each other's capacity to innovate and promote CE practices at the local level and began exploring how to scale impact at higher levels.

*3.2. City-level: Farmers Markets*

3.2.1. Focus 3: Contextual Background and Problem Framing

Farmers markets provide a platform for farmers to sell their produce directly to consumers. Over time, they have expanded from being a point of sale for farmers to providing novel "farm to fork" food for wealthier consumers and addressing food security and access in low-income communities [56,57]. Most recently, they have also started to engage in "shop local" movements to support small businesses, as well as community building through various complementary programming.

Farmers markets can be characterized as goal-oriented, flexible platforms situated across cities. The POEMS design framework (people, object, environment, messages, and services) defines people (market managers and cleaning staff), objects (wayfinding posters, bins, posters, flyers, tends, etc.), environments (physical space, parking, etc.), messages (information regarding vendors and their produce, incentives to buy local and fresh food, food recipes, etc.), and services (cooking classes, kids tables, prepared food and beverages, live music, etc.) [47,58]. As situated platforms, farmers markets enable the allocation and circulation of various types of resources in urban environments by integrating different production and consumption systems (finance, waste collection, mobility, regulatory, entertainment, food, fashion, health, etc.) that shape and condition various interactions between farmers, local businesses, residents, nonprofits, and government organizations [59]. Yet without proper infrastructures to integrate their efforts into other movements, farmers markets are limited in leading large-scale changes, such as those required to implement CE practices citywide.

Farmers markets hold immense potential to demonstrate, promote, and engage urban businesses and residents with CE practices, as they collectively attract hundreds of thousands of visitors each year in Chicago. Yet there are many barriers, including lack of adequate funding, staffing, information, energy, and waste diversion infrastructure, which make it difficult to implement CE beyond market-level initiatives. Grant funding is one of the chief determinants of what activities and programming market managers can develop and incorporate. Although farmers markets depend upon a large number of interactions between diverse sets of agents, they are isolated within their geographic boundaries. As such, each of these interactions, including among market managers across the city, holds an opportunity for education and action around CE.

### 3.2.2. Investigating Opportunities for Scaling Niche-Level CE Success to City-Level Impact

The research team (A.N., E.L., and interns from PC) applied the innovation lenses framework to identify how activities performed by PC staff activated and mobilized the eight capitals and to create new strategies for increasing PC's capacity to lead research and innovation activities that scaled the impact of their local CE work.

Given PC's development of a farmers market at The Plant and burgeoning collaborations with other markets throughout the city, the researchers focused on the unrealized value of farmers markets to advance CE in the City of Chicago and explored them as urban infrastructures to expand and replicate local CE practices [60]. Farmers markets have increasingly aimed to support local businesses while also attempting to tackle food insecurity and malnutrition [61]. The challenge in Chicago of how local CE practices should be incorporated into the rules, regulations, and daily operations of the market itself, has yet to be uncovered. Therefore, the focal question for this phase of the research was: What hard and soft infrastructural changes could enable farmers market managers to advance local circular economy practices in Chicago?

### 3.2.3. Application of the Innovation Lenses Framework in Farmers Markets in Chicago

While farmers markets have been positioned as flexible platforms capable of changing and adapting to address social needs and consumer demands regarding larger societal movements, they operate in isolation. Without proper soft infrastructures to integrate their efforts into other movements, farmers markets are limited in leading large-scale changes, such as those required to implement CE practices citywide. As a result, researchers investigated situations within which farmers markets enabled the integration of multiple systems (e.g., food, mobility, education, health, economy, natural) and supported multiple interactions among diverse sets of agents (e.g., customers, vendors, market managers, hosts, volunteers). To do so, researchers considered each farmers market as a system in itself and conducted ethnographic research on both users and infrastructures through the innovation lenses framework. Table 2 provides a set of questions that guided this phase of research, and examples of the infrastructures, stocks, and flows of different types of resources across them.

PC's staff led most of the activities, such as conducting a literature review on how farmers markets have gone through iterations over time, exploring their contribution, evolution, and adaptation within urban environments during the 19th and 20th centuries. Together, the research team visited different farmers markets, grocery stores, restaurants, urban farms, community centers, and research institutions, and engaged with local vendors, farmers market managers, customers, and peer organizations for participant observations and interviews. At every opportunity, researchers sought to have both formal and informal conversations with diverse agents, including vendors, visitors, farmers, etc., about their personal and professional experiences in farmers markets, and the infrastructure supporting their activities around Chicago.

Upon identifying a variety of resources and assets within farmers markets, the team utilized the anatomy of infrastructures tool (Figure 3) as a structure to organize and make sense of the data gathered. This tool combines principles of multi-level systems mapping, the POEMS design framework, and the innovation lenses framework. It illustrates (1) how different types of resources are flowing through existing offerings; (2) the actionable properties these offerings currently afford users based on the access to specific resources (e.g., affordances); (3) the impacts they generate at different levels considering resource flows and allocation; and (4) their relationships with the main goals.

**Table 2.** Infrastructures, stocks, and flows of resources in farmers markets across Chicago.

| Capital | Dimension | Stocks | | Flows | |
|---|---|---|---|---|---|
| | | Guiding Question | Examples | Guiding Question | Examples |
| **Human** | Knowledge | Whose knowledge and labor are considered? | Market managers knowledge of regulations, advertising, and vendors knowledge of produce and goods | How and where is knowledge being created, and for whom? | Market managers analyze data and activities of markets and act accordingly |
| | Well-being | How is the capacity of individuals to perform defined? | Individuals' health, education | What activities maintain or enhance individuals' capacity to perform? | Cooking classes, healthcare check ups |
| **Social** | Professional | Who is considered a partner in farmers markets? What is the nature of the partnership or affiliation? | Suppliers, market vendors, city officials, volunteers | How and where are partnerships being formed? | Outreach of market managers, vendor applications, networking events across the city, vendor training |
| | Personal | What informal ties exist within current operations? | Relations among managers, vendors, and customers help them to participate in the market | How and where are activities supporting informal gatherings happening? | Informal social gatherings in designated common areas among managers, vendors, and customers |
| **Cultural** | Local | What are the local traditions and cultural heritage farmers markets rely upon, and what values and beliefs they sustain? | Local food producers, activists of local and organic movements, language and vocabulary used to communicate with visitors | How and where are the cultural practices and values manifested? | Selection of farmers/vendors/ activities presented at market |
| | Global | What global elements and practices have been incorporated? | Organic certification/standards, variety of attractions beyond food (e.g., health, music, art, dance), bilingual communications | How and where are new global practices being incorporated? | Market managers host diverse activities, media messaging about food safety and health, local grocery stores pose competition |
| **Political** | Regulations | What are the local, state, and federal policies influencing decisions in farmers markets? | City regulations, state/federal food safety rules, funding available for different types of markets, food assistance programs | How and where are policies being enforced or changed? | Market managers have to follow food safety compliance, vendors, and double-value data collection, waste management |
| | Norms | What is the power structure within current operations? | Market owners/managers make decisions for all participants, vendors, and consumers | How and where are decisions being made, or power shifts taking place? | Decisions are typically made by owners/managers, outside of the market, and presented to vendors prior to the event |

**Table 2.** *Cont.*

| Capital | Dimension | Stocks | | Flows | |
|---|---|---|---|---|---|
| | | Guiding Question | Examples | Guiding Question | Examples |
| **Natural** | Fauna and Flora | What is the composition of flora and fauna species supporting the farmers markets? | Vegetables and animal products sold, organic and conventionally produced within and outside the city | How are species growth rates affected by the market activities? | Farming practices, organic and conventional, tend to disrupt the ecosystems in which they exist |
| | Life support systems | What are the energy, materials, and services provided by nature to farmers markets? | Soil for growing produce to be consumed by people or other animals | How and where are energy and nutrients being extracted and regenerated? | Extraction of resources happen within and outside the city, most markets outsource their waste management services |
| **Financial** | Services | What institutions provide financial services to agents participating in the markets? | Loans: banks, credit unions; Grants: federal, state, and local governments, foundations; Payments: consumer income, government food assistance | How and where are financial services being provided? | Formal financing of vendor operations; on-site ATMs, SNAP program, double-value services |
| | Money | What is the institutional structure defining value? | Financial institutions, philanthropic foundations, market competition | How and where are monetary flows occurring? | Outside market: grants, expenses; Inside market: product sale transactions |
| **Manufactured** | Infrastructure | What is the physical infrastructure available and its condition? | Transportation networks, automobiles, market-owned space furniture, building and city utilities | How and where is physical infrastructure being used and enhanced? | Location of the market determines accessibility and space occupation; seasonal markets may occupy different spaces |
| | Products and Services | What products, byproducts, and services support activities? | Packaging, marketing materials, educational activities, kids table | How and where are products and services being produced and consumed? | Packaging typically produced elsewhere and discarded at consumers' homes. |
| **Digital** | Infrastructure | What are the digital infrastructures available? | Points of sale systems, social media, computers | How and where are digital infrastructure and data used and enhanced? | Data analyzed to gain additional funding |
| | Data and Information | What data and information supports activities? | Quantified attendance, sales data | How and where are data collected and managed? | Managers and vendors collect data on sales, attendance, promotions |

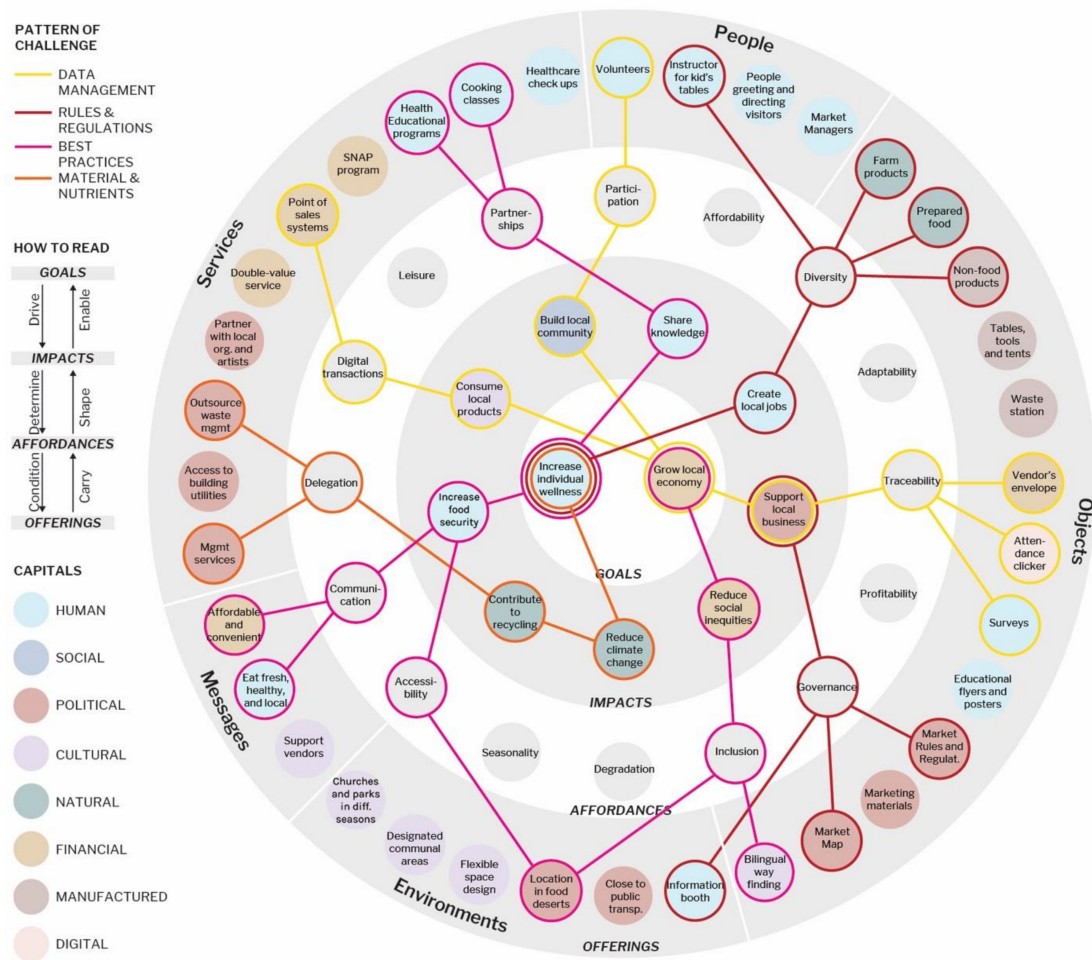

**Figure 3.** Anatomy of infrastructures underlying farmers markets in the City of Chicago.

The anatomy of infrastructures tool supports the investigation of the overall alignment between the offerings of a system (outer circle) and the intended goals of an infrastructure, situated in the center of the diagram. It considers that resources flow through infrastructures considering actionable properties that their offerings bring to reality, and the impacts users generate given the intended goals. This tool follows the subsequent logic: the goals of man-made infrastructures (center of the diagram) determine the intended and unintended impacts that individuals and organizations create in the broader context in which they exist when they mobilize different resources. These impacts are determined by the possible actions that users can take given the offerings available (outer circles), thereby suggesting how individuals and organizations may use or leverage them through their activities. The correlation between offerings, affordances, impacts, and goals is helpful to understand the current relationships within the hard and soft dimensions of the infrastructure being analyzed, and how these four elements can be integrated into new system interventions.

The team used this tool to understand the infrastructural complexity in advancing CE in farmers markets, and how interventions should consider the integration of infrastructures present in other systems, beyond the farmers market system. For example, the team learned that a set of vendors developed the capacity, relationships, and reputation to sell their products in different markets across the city, but their footprint leaves little room for local or neighborhood businesses to compete within these spaces. Even though diversity is an affordance of current infrastructures that enables a variety of offerings to come to life in each market, the same property is not being leveraged for addressing challenges underlying social inequalities, especially those related to the diversity of vendors across markets.

### 3.2.4. Patterns of Challenges in Mobilizing Resources

The application of new frameworks and tools led researchers and managers to identify four patterns of challenges related to allocating and mobilizing different types of capital through the farmers market infrastructures.

- Digital and financial capital—managing diverse sets of data: Integrating data from multiple sources and transforming the results into meaningful information requires new competencies within current managerial practices. Farmers markets' managers face the same challenges that managers in large organizations face: what data should be collected? Which sources should be integrated, and to what end? Yet, because farmers markets are usually run by small organizations with resource constraints, they had to develop the minimum necessary set of competencies to create the intended values for specific user groups. For this reason, they often do not have the capabilities to integrate environmental- or social-related data into their operations. This focus revealed an opportunity for collaboration around benchmarking and data sharing to communicate local circular economy practices with each other and with the City government.

- Political and cultural capital—updating rules and regulations: Market managers tend to align their practices and regulations with the larger goals of the City of Chicago programs. Markets must comply with a wide range of rules and regulations, but existing CE practices have emerged organically. Thus, where CE practices exist, they are defined by the vendor's interest and are not explicitly encouraged by the City government. Through this work, the managers are exploring how they can shape new market regulations to encourage the adoption of local circular economy practices by vendors and customers.

- Human and social capital—sharing best practices: Several CE practices have taken root in markets across the city, but opportunities to accelerate and increase impact remain untapped. Chicago's market managers currently share best practices with each other through informal relationships and infrequent gatherings. Without a formal infrastructure to create and sustain more open and inclusive mechanisms to share their learnings and experiences, many managers go through the same struggle at different times. Consequently, the overall farmers market operations become inefficient in resource allocation. This realization led managers to create a systematic way to learn from each other, employ proven successful practices at their markets, and advance the local circular economy.

- Natural and manufactured capital—managing materials and nutrients: Implementing a materials and nutrient management system based on CE practices can significantly increase the diversion of recyclable and biodegradable materials from landfills. In the past five years, a handful of markets have hired small, local food scrap haulers to handle their organic waste, but many markets identified lack of budget, infrastructure, knowledge, and available local services as barriers to diverting materials and nutrients from landfills. Since the majority of waste generated in farmers markets is biodegradable, the ability to implement a system that properly manages materials and nutrients is highly dependent on the market's host site and the priorities of the market's host organization. This insight led to the consideration of how managers could collaborate to create an affordable and effective system to divert materials and nutrients away from landfills.

### 3.2.5. CE Interventions

The researchers hosted a design-led workshop at the IIT Institute of Design for market managers, representatives of the City of Chicago, and other stakeholders working on CE initiatives from the private sector and non-governmental organizations (NGOs). The goal was to co-create alternative paths to advance local CE through their markets. The workshop represented an action situation (see [18]) necessary to advance CE practices in complex, open-ended projects involving multi-stakeholders, such as urban farmers markets. Each participant received a contextual report created by the researchers indicating the approach to the research and the four common challenges that were uncovered. After

validating interpretations about these four challenges, participants divided themselves into small groups, each responsible for one challenge, and considered principles of transparency, diversity, and inclusion as underlying criteria for intervening in current flows and allocation of resources. Participants agreed upon (1) a common goal (advance local circular economy in Chicago through farmers markets), (2) a set of challenges, and (3) the criteria for intervention (principles for local CE) to co-define principles for future engagements and explore new competencies that market managers needed to enable local CE. These priorities and competencies are education (knowledge dissemination), facilitation (CE-oriented interactions to sustain engagement with vendors and customers), collaboration (organize and intentionally support one another), and coordination (ensure that collaboration leads to actions). Combined, these four competencies present opportunity areas for intervention and impact related to building soft infrastructures for the CE at the city-level (see Figure 4).

The understanding and implementation of CE practices at farmers markets require market managers to be connected and communicate regularly. This project served as a starting point for a series of collective efforts to increase the adoption of CE practices currently being led by a coalition of farmers market managers. A research report was co-created and is currently being used by PC staff and other market managers to continue to form new engagements with various stakeholders involved in farmers markets activities [60]. Stakeholder group activities currently include advising the City of Chicago on market regulations, changing the rules and regulations of individual farmers markets programs, maintaining a digital platform for market managers to communicate, and building vendor's capacity to engage in CE practices, among others. This highlights the opportunity for scaling the impact of meso-level successes, by pivoting through strategic alliances, systems understanding, and infrastructuring co-design practices with lead agents and partners.

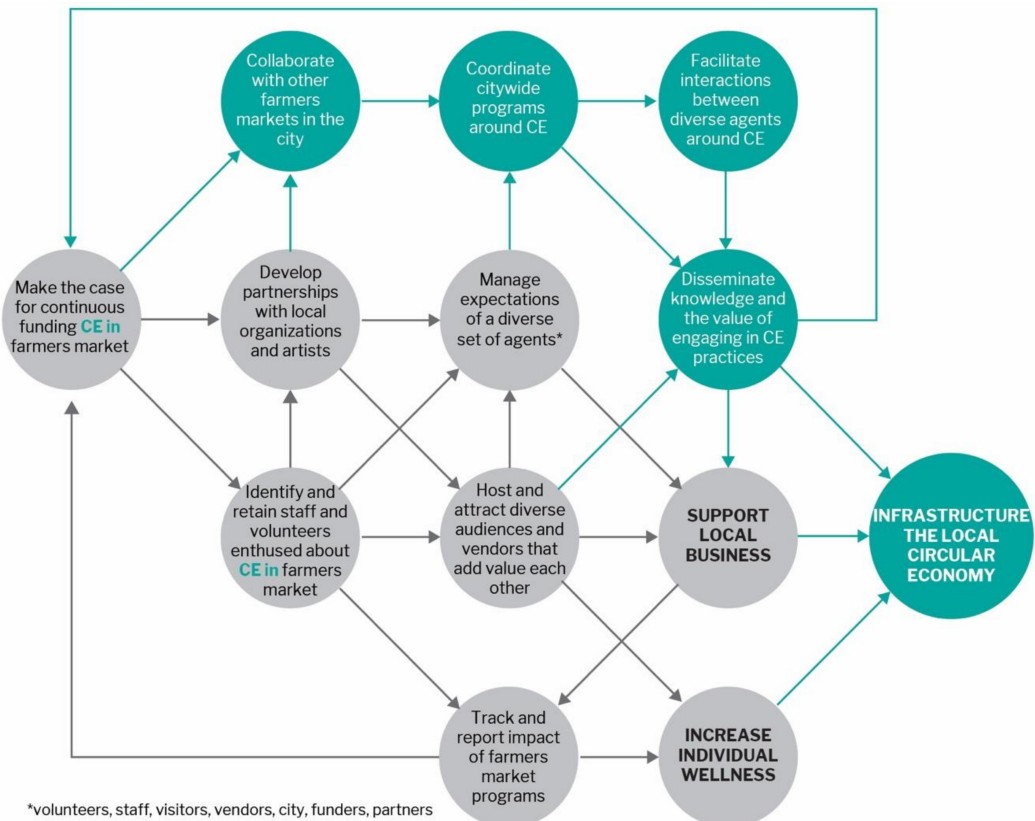

**Note:** Grey elements illustrate existing competencies needed to bring a farmers market to life, and their interdependencies (arrows). Green elements (arrows, texts, and circles) situate the changes and the four new competencies identified by researchers and participants.

**Figure 4.** Competencies needed for infrastructuring the local CE through farmers markets.

### 3.2.6. Four I's Model

Combined, the experiences at the facility-level (The Plant) and the city-level (farmers markets in Chicago) culminated in the "Four I's" model, which presents four intervention strategies needed in any process of "infrastructuring" the CE practices in SETS, regardless of its geographic boundary. The "Four I's" model consists of: (1) interconnectivity between the organizational levels of systems; (2) integration of the social, ecological, and technical systems shaping the conditions in place; (3) interactions of the agents defining the dynamics of these systems; and (4) iteration of design interventions over time [62]. The Four I's model ultimately presents four general design strategies for intervening in complex CE challenges, such as in urban environments. For each one of these strategies, researchers are exploring a specific tool that can support and advance design practices in these complex spaces.

#### Interconnectivity of Organizational Levels

People experience different conditions of SETS and develop unique knowledge about them depending on the level within which they are embedded (macro, meso, micro). Unlike embedded (operational) and explicit (codified) knowledge, "tacit knowledge can neither be explained in terms of rational decision-making nor be summarized easily in quantitative terms" [63] (p. 55). This challenges traditional approaches to expertise and requires different stakeholders to understand the value of daily life experiences that manifests itself as tacit knowledge, to inform change. Thus, in addition to capturing the different types of knowledge about the conditions in each one of these levels, individuals and organizations need new approaches that recognize and incorporate considerations of how the interdependency among them is shaping the dynamics of SETS. Although activities at each level need certain autonomy to increase efficiency and effectiveness, they also need to be connected and integrated with the choices and activities happening at the other levels. Without proper alignment, the chances of unintended consequences might increase because a choice made in one level invariably will be made based on the unrealistic assumption that the other levels will support or are capable of adapting accordingly.

#### Integration of Multiple Systems

A system is a collection or set of interconnected parts, usually delimited by some type of spatial or temporal boundaries. Systems' boundaries can be outlined based on the structure, the functionality, the nature, and/or the intended goal of the analysis and design [64]. As individuals and organizations focused on creating new CE practices in cities tend to interpret systems' boundaries of their innovation processes based on the specialized practices of their own industry or sector, they create artificial boundaries for interventions that are defined by their focal idea, not by the natural or existing limits of the system. Yet, infrastructural interventions in urban areas result from individual activities and organizational practices that activate and mobilize various types of resources currently distributed across multiple systems (e.g., energy, knowledge, water, money, power, etc.). Thus, when these agents take traditional approaches that consider one system at a time, they overlook the extent through which infrastructural interventions are conditioned by, and dependent on, intersecting systems that shape everyday life [65].

#### Interactions of Diverse Sets of Agents

The interaction of diverse agents can bring distinctive perspectives to framing problems and developing solutions to CE challenges in cities. Agents can be human or non-human, including components of technology (e.g., portable machines, digital platforms, organizations, products, etc.) or biological (living creatures, etc.). An interaction in SETS occurs when any agent affects the other. Interactions can be defined as symbiotic relationships between and among social, technical, and ecological components. They are dynamic relations between parts and wholes in systems and can be transactional (e.g., a single purchase from a vendor at a farmers market) or defined by a temporal

pattern (e.g., increase of visitors at The Plant during wintertime). Redesigning the flows of political capital within various systems that shape local dynamics should underlie efforts to create infrastructural interventions. Currently, this resource is unevenly distributed and hoarded by certain agents within SETS. Such conditions create unsustainable power dynamics among agents and influence the outcomes of infrastructural interventions. Boonstra defines power as "a (human) capacity to act in social and ecological conditions" [66] (p. 1). Thus, understanding the role of power in shaping contemporary socio-technical and socio-ecological interactions in urban territories is critical to map and intervene in current urban dynamics. As noted by Geels, existing institutional arrangements operate within certain power dynamics, and to intervene in them, individuals and organizations that seek to create urban CE practices need to understand who has the power to enable or inhibit large-scale transformations, including how the relationships among and between these two groups are sustained [67].

Iterations over Time

Having considered the dynamic interactions of diverse sets of agents as the third attribute of infrastructural intervention, this last part addresses the iterative nature of CE interventions. Infrastructural interventions "are always and unavoidably situated within, and part of the sedimentation of material arrangements, themselves linked to a persistently dynamic profile of activities and practices" [65] (p. 164). Infrastructuring the CE demands recognition that infrastructures are path-dependent and impermanent because of the variation of the activities they enable. The dynamic nature of everyday individual activities and organizational practices suggests that intervening in existing infrastructures to support local CE demands agents involved in these processes to work in successive intervention processes, as the interactions between diverse sets of agents utilizing these infrastructures will necessarily change over time.

Cities, as SETS, are shaped by interdependent infrastructures that produce synergies due to complementary functions they perform in activating and mobilizing different types of resources. Such interdependence will likely support the co-evolution of both problems and solutions to city dwellers, organizations, and ecosystems within the urban territory. Thus, infrastructural interventions are dependent on how different resources distributed across social, technical, and ecological systems can be activated and mobilized so as to serve one or more SET demands (e.g., Bubbly Dynamics repurposing a building, while contributing to local economic development). This work suggests that diverse agents driving CE change in cities need to adopt more pliant approaches that can enable them to continuously engage and involve diverse voices representatives of the plurality of residents and local business in both processes of problem framing and solution finding, as well as in the resulting pathways that lead to creating and implementing infrastructural interventions for local CE practices.

## 4. Discussion

A significant gap remains to connect conceptual and ideological discourses of CE practices to the pragmatism required to intervene in complex situations that need reconsideration of the allocation and circulation of different types of resources, so they are fit to the sustainability of the SETS of interest. Transitions in SETS, such as cities, require knowledge integration from both multiple disciplines and diverse agents distributed across the different levels of these systems. As knowledge is dispersed in multiple forms and agents shape the dynamics in SETS, new approaches are required to not only increase impact in each one of these systems but also to carve new opportunities and explore emerging possibilities. This work focused on creating innovative infrastructural interventions centered on overcoming complex socio-ecological-technical challenges. More specifically, it focused on expanding CE practices to contribute to the sustainability of the systems within which they exist and advancing expertise in providing alternative, regenerative approaches to scale impact from niche-level successes to higher levels in SETS.

In the case of both the facility- and city-level, the opportunity relied on the creation of new knowledge infrastructures, or "robust networks of people, artifacts, and institutions that generate,

share, and maintain specific knowledge about the human and natural worlds [68] (p. 5). Here, knowledge infrastructures are recognized as adaptive and involving a continuous flow of information because (1) individual elements are constantly changing, leaving, or even being introduced, and (2) "knowledge is perpetually in motion" [64] (p. 6), meaning that the definition of the known is constantly changing either by novel questions, redefinitions, or incorporations of novel perspectives, which was the case of the CE paradigm. Such fluid conditions frame knowledge as a resource flowing through the interactions between different agents in SETS. As Edwards and colleagues argued, "the current situation for knowledge infrastructures is characterized by rapid change in existing systems and introduction of new ones, resulting in severe strains on those elements with the greatest inertia" [68] (p. 5). Thus, if individuals and organizations interested in enabling CE practices intend to capture these resources and embody them into new interventions, they will have to develop new mechanisms to integrate knowledge that is not only pulled from different domains (e.g., industrial ecology, socio-technical, design, and socio-ecological systems theory) but also distributed among different agents (residents, local institutions, researchers, local businesses, investors, government agencies, etc.), considering how they are interacting on multiple levels of the SETS of interest. By doing so, individuals and organizations may be better equipped to design alternative infrastructures capable of confronting complex socio-ecological-technical challenges preventing the wide adoption of CE practices within urban environments.

As urban change makers, researchers working with individuals and organizations embedded in cities leveraged the innovation lenses framework and the Four I's model to explore infrastructure within The Plant and in farmers markets across Chicago. Both were approached as junctions of social, ecological, and technical systems enabling the allocation and movement of multiple types of resources. Upon completing the first comprehensive material flow analysis of The Plant, our research focused on co-creating a sustainability measurement framework to support The Plant's tenants in their engagement with CE activities. By applying prototyping methods as means for infrastructuring change through PAR, the team uncovered the need, and opportunity, to broaden participation and involvement in determining and shaping CE activities both within and beyond the facility (the niche), and to consider the variety of perspectives (Four I's model) and resources (innovation lenses framework) that shape sustainability and equity within SETS dynamics. As a response, Plant Chicago repositioned itself as a convener for local circular economy practices, facilitating niche-to-regime transitions towards circularity across the city, in this case, through farmers markets.

This research project not only helped build design capacity among PC staff, increasing their ability to embed principles of sustainability and equity into their innovation processes towards CE, but it also increased their confidence in tackling complex projects at a broader scale, beyond the facility boundaries. In the process of doing so, PC transitioned from an NGO that promoted CE through research and development at The Plant to one cultivating local circular economies across the City of Chicago by convening diverse perspectives that would not otherwise have a forum in the circular economy space. Researchers, on the other hand, benefited from social connections and political access to various communities, and from the knowledge and experience of diverse agents embedded in these communities as they had deep expertise about existing infrastructural challenges, as well as historical and cultural patterns that continue to prevent CE to be actualized. Such contributions were keen to advance on new approaches, combining frameworks, methods, models, and tools that can facilitate and contribute to the sustainability and equity of SETS dynamics through CE practices. Other agents directly or indirectly involved in this research benefited from the activism and leadership of PC to intervene in the flows of resources through infrastructural interventions.

The researchers, PC staff, and market managers explored farmers markets as citywide infrastructures to promote and enable local CE practices, rather than isolated platforms. Together with other market managers, the research team realized that by incentivizing changes in the business practices of small companies involved in farmers markets, vendors will likely change their practices elsewhere, consequently creating a network effect across the city with the farmers market as a leverage

point for change. As urban infrastructures are (re)oriented towards CE, farmers markets in Chicago present high potential to become the means through which other agents learn about and can practice CE. Currently, they are not only becoming regulated by CE principles, but also provide incentives for the adoption of CE practices through various activities (e.g., capacity building of vendors, periodic meetings of market managers), aligned messages (e.g., posters informing opportunities and decisions around CE, more inclusive and diverse messaging to reflect the population it serves, etc.), and new offerings (e.g., new waste stations, compost services, advice and recommendations for policy change in the City of Chicago, among others). By relying on the concept of infrastructuring, market managers are creating new CE-focused programs that increase interactions between small local businesses, while also forming a local network centered on CE. In doing so, they are connecting data, information, and strategies from attempts in each market, sharing with each other, and co-creating and prototyping alternative interventions based on the resources available and on the opportunities to promote well-coordinated, citywide change towards CE.

## 5. Conclusions

The application of the innovation lenses framework and Four I's model can help agents in public, private, and civil society to consider how infrastructural interventions activate, mobilize, and are conditioned by the flow and allocation of various types of resources, the interconnectivity of different organizational levels, the intersection of social, ecological, and technical systems, the interactions of diverse sets of agents, and their iterations over time. Urban infrastructures are the means through which resources flow in a system, given a specific goal, traditionally to support economic growth through the lenses of technical systems. Today, there is increasing recognition that they have to be adapted and used to support more sustainable and equitable outcomes in urban environments; but to do so, urban change makers must develop their expertise in creating the infrastructural interventions affecting the circulation of different types of resources, and contribute to the fitness of humans and non-human agents' interactions happening within and across systems levels.

Exploring infrastructure from a multi-level, relational perspective unlocks new opportunities for situated urban interventions to consider how the relationships between different agents embedded in these situations are the means through which different types of resources are activated and mobilized within a city. Expanding the traditional boundaries of infrastructural problems to include dynamic interactions of diverse agents underpinning resources flow also enables urban change makers to better understand how these dynamics in turn shape and condition situated interventions. When explored as elements of both socio-ecological and socio-technical systems, infrastructural interventions for CE in urban settings become the nexus through which different types of resources are combined to generate transformational change towards sustainability and equity.

**Author Contributions:** Conceptualization, A.N., W.A. and C.T.; methodology, A.N., W.A., C.T., E.L. and J.P.; formal analysis, A.N.; investigation, A.N. and E.L.; resources, A.N., W.A., C.T., E.L. and J.P.; data curation, A.N.; writing—original draft preparation, A.N. and W.A.; writing—review and editing, A.N., W.A., C.T., E.L. and J.P.; visualization, A.N.; supervision, W.A. and C.T.; project administration, A.N., W.A. and C.T.; funding acquisition, W.A., C.T. and A.N. All authors have read and agreed to the published version of the manuscript.

**Funding:** This research was funded by Illinois Institute of Technology's Education and Research Initiative Fund, grant number 17-0344 and the APC was funded by MDPI.

**Acknowledgments:** The authors wish to thank all the stakeholders at The Plant, in particular John Edel, Carolee Kokola, and Alex Enarson of Bubbly Dynamics, Rosanna Lloyd and the employees of Just Ice, Art Jackson and the employees of Pleasant House Bakery, Adam Pollack and the employees of Closed Loop Farms, Alex Poltorak and the employees of Urban Canopy, Kaley Donewald from Secred Serve, Brian Taylor and Charla from Whiner Brewery, Ria Neri from Whiner Brewery and Four Letter Word Coffee, Matt Lancor from Kombutchade, for generously sharing their time and knowledge. We also wish to thank the Spring 2017 Sustainability Management students: Roshni Lad, John Lerczak, Megan Mckitterick, Flavio Santos Urteaga, and Tian Wen, and Plant Chicago Summer 2017 interns: Tommy Straus and Tyler Washington, who worked with us to interview stakeholders, collect and organize data.

**Conflicts of Interest:** E.L. and J.P. are employed by Plant Chicago, the non-government organization featured in the manuscript. A.N. and W.A. are members of Plant Chicago's Circular Economy Advisory Board.

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
