# Peer review of "Infrastructuring the Circular Economy"

_energies, doi:10.3390/en13071805_

Round 1

Reviewer 1 Report

see the attached PDF with indications

the paper must be reorganised in order to present the research and the results followed by conclusion and future developments

- there are to much descriptive parts and a big mix of the chapter contents 

Reviewer 2 Report

This manuscript presents a framework and case study for bridging the gap between localized CE implementation and their potential effects at a larger urban scale. The study is interesting and timely, and provides meaningful insights into a case study dealing with urban food systems. 

My main concern is the structure and goal of the article. There seems to be a mix of methods and case study results throughout the paper, so I would advice the authors to review/reconsider some of these comments. Hopefully they are helpful:

  • The introduction is nicely written. It wasn't until I read the results, however, that I realized that this study revolves around a case study on food production and farmers markets. The introduction describes the challenges of urban food distribution as an example, but not as a way of walking the reader through the rationale of the study. I get the impression that the case study is particularly relevant for the analysis and it seems to be the core of the article. I would advice the authors to revisit their goals and research questions. As of now, the goal is very general in the context of the CE in cities, but then the results and conclusions present a specific case. 
  • Section 1.1. is very self-explanatory (thank you for that!). It is a bit out place, it seems. Would it be part of the methods? This section appears right after the goal formulation and is breaking the flow between goal and methods. 
  • I wonder, however, what the main goal is. Is it to present a framework or to address a case study through a framework. I found your figures in the results extremely interesting and I would have liked to know more about them, but the results focus more on the "how" instead of the "what". I would suggest a clear separation between methods and results. 
  • I can see that your methods are quite complex and maybe that is why it is complicated to follow the results. I would personally appreaciate a figure describing your methods that can then walk the reader through your results. This would be extremely helpful. The methods would then include the description of the case study. I don't think that the case study itself is a result.
  • This figure could also include the changes in scale. The examples provided for micro to macro scales were helpful, but then other concepts appear and they are no longer so apparent. 
  • Please make sure to highlight the relevance or the representativeness of this case study when addressing the circular economy and your framework. The paragraph in lines 457-471 seems to serve this purpose, for example. 
  • Another comment on this paragraph (lines 457-471) is about the question posed at the end. Is that the paper's research question or a subquestion? Perhaps it will be clearer once you define your methodology. If several questions are asked throughout the research process, you could create a flow diagram indicating the steps you followed and the questions needed in each case.
  • If the theoretical framework is also a product of your research, my next question is: which of them? You first present your combination of methods, then there's the Innovation Lenses framework, and finally the Four I's model. Which one is the product and which ones are the methods? This was a bit confusing to me.

Additional clarification questions:

  • What is the relationship between the prototypes and your combination of methods? The prototypes are only mentioned in the methods.
  • Line 256: when is the Whole View Model used? 
  • Line 260: is the network analysis a step to generate a systems dynamics map? I cannot seem to find the network analysis in the results.
  • The Plant is first mentioned in line 306 but it is difficult to relate to the previous paragraph. Please reformulate.
  • Is there a complete list of CE strategies discussed or implemented at The Plant? I have the impression that the strategies are not presented at all. Or have I missed something? Material reuse and recycling are mentioned in the methods, but this is very general.
  • Line 437: I believe the 8 capitals would also be a method or concept, wouldn't they?
  • I guess that the Anatomy of Infrastructures is the system dynamic map, isn't it? Somehow the terminology is a bit mixed up. Are the goals, impacts, affordances and offerings part of the existing method or did the authors come up with these categories?
  • To create this anatomy map, did the authors use the results generated through the questions in table 2? These two parts are a bit disconnected. 
  • Are the connections between two circles directional? It seems they are, based on the legend, but I can't seem to find an arrow pointing to one of the circles. Should they? Or I am reading this incorrectly? 

Round 2

Reviewer 1 Report

the authors didn't upload the paper with the requested modification visible, and for me this is not correct

The reviewers spend time to read and give indications and some authors decide to pass and try to be smarter than the reviewers 

Reviewer 2 Report

The authors addressed most of my comments and the article now points to the creation of the two frameworks more clearly. I recommend this manuscript for publication

Round 3

Reviewer 1 Report

the paper was improved and can be accepted ( the subject could be of interest), it seems that the authors however chose to make simplified changes.